

# Newly incident cannabis use in the United States, 2002–2011: a regional and state level benchmark

Jacob P. Leinweber[1], Hui G. Cheng[1], Catalina Lopez-Quintero[2] and James C. Anthony[1]

[1] Department of Epidemiology & Biostatistics, Michigan State University, East Lansing, MI, United States of America
[2] Substance Use and HIV Neuropsychology Lab, Center for Children and Families, Miami, FL, United States of America

## ABSTRACT

**Background.** Cannabis use and cannabis regulatory policies recently re-surfaced as noteworthy global research and social media topics, including claims that Mexicans have been sending cannabis and other drug supplies through a porous border into the United States. These circumstances prompted us to conduct an epidemiological test of whether the states bordering Mexico had exceptionally large cannabis incidence rates for 2002–2011. The resulting range of cannabis incidence rates disclosed here can serve as 2002–2011 benchmark values against which estimates from later years can be compared.

**Methods.** The population under study is 12-to-24-year-old non-institutionalized civilian community residents of the US, sampled and assessed with confidential audio computer-assisted self-interviews (ACASI) during National Surveys on Drug Use and Health, 2002–2011 (aggregate $n \sim 420,000$) for which public use datasets were available. We estimated state-specific cannabis incidence rates based on independent replication sample surveys across these years, and derived meta-analysis estimates for 10 pre-specified regions, including the Mexico border region.

**Results.** From meta-analysis, the estimated annual incidence rate for cannabis use in the Mexico Border Region is 5% (95% CI [4%–7%]), which is not an exceptional value relative to the overall US estimate of 6% (95% CI [5%–6%]). Geographically quite distant from Mexico and from states of the western US with liberalized cannabis policies, the North Atlantic Region population has the numerically largest incidence estimate at 7% (95% CI [6%–8%]), while the Gulf of Mexico Border Region population has the lowest incidence rate at 5% (95% CI [4%–6%]). Within the set of state-specific estimates, Vermont's and Utah's populations have the largest and smallest incidence rates, respectively (VT: 9%; 95% CI [8%–10%]; UT: 3%; 95% CI [3%–4%]).

**Discussion.** Based on this study's estimates, among 12-to-24-year-old US community residents, an estimated 6% start to use cannabis each year (roughly one in 16). Relatively minor variation in region-wise and state-level estimates is seen, although Vermont and Utah might be exceptional. As of 2011, proximity to Mexico, to Canada, and to the western states with liberalized policies apparently has induced little variation in cannabis incidence rates. Our primary intent was to create a set of benchmark estimates for state-specific and region-specific population incidence rates for cannabis use, using meta-analysis based on independent US survey replications. Public health officials and

Corresponding author
James C. Anthony,
janthony@msu.edu

policy analysts now can use these benchmark estimates from 2002–2011 for planning, and in comparisons with newer estimates.

## INTRODUCTION

Drug supply and availability occupy central positions among environmental conditions and processes that account for risk of becoming a drug user, with a potentially cascading influence on transitions from newly incident drug use toward associated problems, such as drug dependence syndromes or addiction states (*Volkow & Li, 2005*). In research on cannabis (marijuana, marihuana), a recent illustration can be seen from evidence on twins and college students born within the United States (US), which supports this view of cannabis availability and 'exposure opportunities' as major environmental influences on becoming a cannabis user, and on occurrence of cannabis problems, once cannabis use starts (*Gillespie, Neale & Kendler, 2009*; *Pinchevsky et al., 2012*).

These ideas about 'drug exposure opportunities' and associated environmental variations emerged from Wade Hampton Frost's early conceptualization of epidemiology as a population science, including his specification, in 1928, of an interacting 'agent-host-environment' triad that now guides public health research generally (*Frost, 1976*). In Frost's triad model, the 'agent' functions as a necessary but not sufficient cause, and motivates epidemiological attention to 'hosts' who live close by environmental 'reservoirs' that support agent viability and propagation—that is, the origin or sources of 'supply' from which agents are conveyed until they make effective contact with susceptible hosts. Hosts living in geographical areas that are distant from an agent's reservoir, with limited or no 'exposure opportunity' for that agent, should have lower incidence rates. These hosts should be less likely to become newly infected or newly incident cases of the disease, whereas those living close by the agent's reservoir might well have more exposure opportunities, as well as associated greater risk for becoming infected. *Wagner & Anthony (2002)* extended Frost's concepts, by analogy, to cannabis and other drug use.

For several reasons, definitive evidence on the 'cannabis reservoir' and associated 'place by place' geographic variations in cannabis availability within the US is in short supply. Drug law enforcement agencies do not yet take a systematic approach to 'controlled buys,' as might be used to discriminate area-specific prices charged at retail versus wholesale levels (*Manski, Pepper & Petrie, 2001*). Furthermore, in contrast to epidemiological research on tobacco cigarettes, studies of internationally regulated drugs of illegal origin do not include a common metric of 'one cigarette' and 'one pack' that can be used to construct pack-years of personal exposure; state and substate jurisdictional tax receipts are not yet useful. For reasons of this type, many drug researchers have raised serious questions about the evidence base required to evaluate the impact of federal or state drug policies regulating drug supply and availability (*Manski, Pepper & Petrie, 2001*). In a more recent critique prepared for the
US federal government and focused on cannabis specifically, scientists working for the RAND Corporation used the phrase 'lacks credibility' when describing the federal approach now used to estimate the amount of cannabis available in US markets (*United States Office of National Drug Control Policy, 2012*).

Notwithstanding critiques along these lines, against a background of increasing domestic crop yield within US borders, it is possible to draw a fairly firm conclusion that Mexico is the country that has been supplying the most cannabis to the US market, by volume. According to a series of World Drug Reports contributed by the United Nations Office on Drugs and Crime, in addition to cannabis trafficking flowing primarily from Mexico into the United States, both Mexico and the US have been the top two countries in cannabis herb seizures since the turn of the 21st century. These two countries alone have made up at least one third of all seizures worldwide (*United Nations Office on Drugs and Crime, 2016*). In contrast, to the extent that available law enforcement data can be trusted for rough comparative purposes, the Canadian supply has been characterized as 'small' (*United States Office of National Drug Control Policy, 2012*).

Figure 1 helps substantiate what now is known about the 'cannabis reservoir' relative to the geography of the United States. It is adapted from an illustration prepared by the US Library of Congress Federal Research Division and National Drug Intelligence Center. It confirms Mexico's position as a major source country in its depiction of both land and sea cross-border trafficking routes for cannabis. Nevertheless, it should be noted that the map leaves out what is presumed to be much more limited trafficking via governmental postal systems and international flights (*United States Library of Congress, 2003*).

Coupled with prevailing theories about importance of cannabis availability, this type of map leaves an impression that cannabis availability, and possibly the incidence rates for becoming a newly incident cannabis user, might be greater for populations living along the US border with Mexico, relative to other regions of the US. Accordingly, we set out to investigate this speculative proposition, with a deliberately naïve advance expectation that the incidence rates for cannabis use might be greater in the states along the US-Mexico border (as depicted West to East in Fig. 1: California, Arizona, New Mexico, and Texas).

Of course, in this work, we had to allow for the possibility that proximity to the cannabis croplands of Mexico might not be the only governing influence. This logic prompted us to set aside California from our originally pre-specified regional grouping of border states, given that the California population has had a tradition of relatively liberalized views about cannabis and other drug use. In addition, California shares a Pacific Ocean coastline with Mexico. Northern California has its own extensive cannabis croplands. Finally, there is an extensive set of roads linking northern California with Oregon across these two western US states with well-documented progressive politics.

In the background, we also had to take into account the possibility that cross-border smugglers might make a better profit by driving their just-smuggled cannabis to a non-border state, in which case street-level retail profit margins for cannabis might lead to greater incidence in non-border states distant from Mexico, such as Nevada, Colorado, Utah, and Oklahoma (*Manski, Pepper & Petrie, 2001*). For reasons of this type, our presentation of state-level estimates in this paper makes it possible for readers to evaluate and re-configure

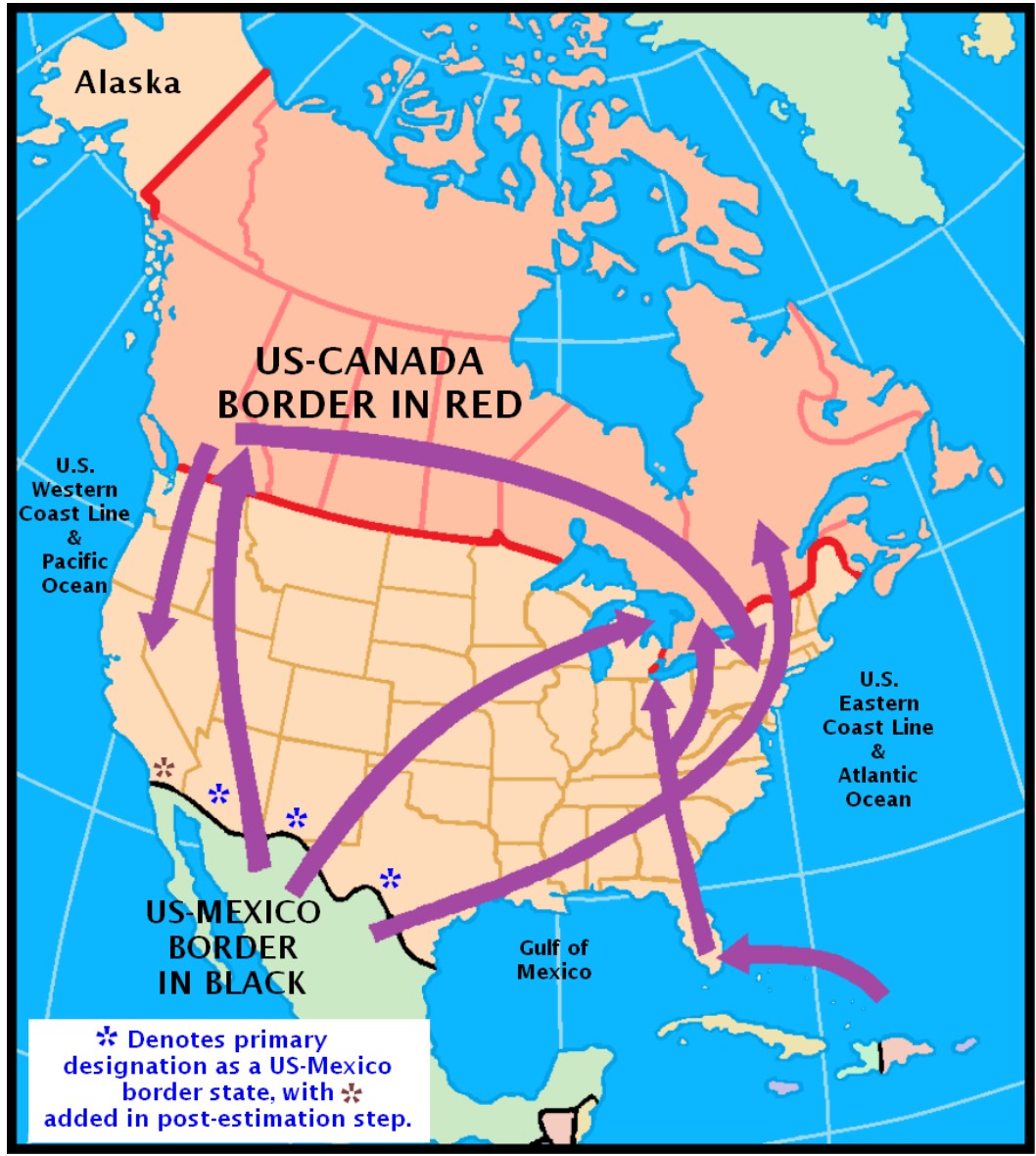

**Figure 1** Reproduction of **Fig. 2** in the Library of Congress report on cannabis (marijuana) availability in the United States (*United States Library of Congress, 2003*).

our regional specifications as a check on what we drew up before any analyses. In addition, the authors can provide CSV and Stata .dta data files with state-level estimates, as well as their standard errors, so that interested readers can study the state-level estimates and can re-configure the regional specifications as they see fit.

In this line of research on cannabis incidence rate estimates, several prior contributions deserve attention, but have not addressed US region-wise or state-specific cannabis incidence rates directly. To illustrate, Rhodes and colleagues (*2003*) provided a fairly comprehensive overview of major determinants of cannabis smoking, over and above the issue of border proximity and area variations in availability, price, and regulations.

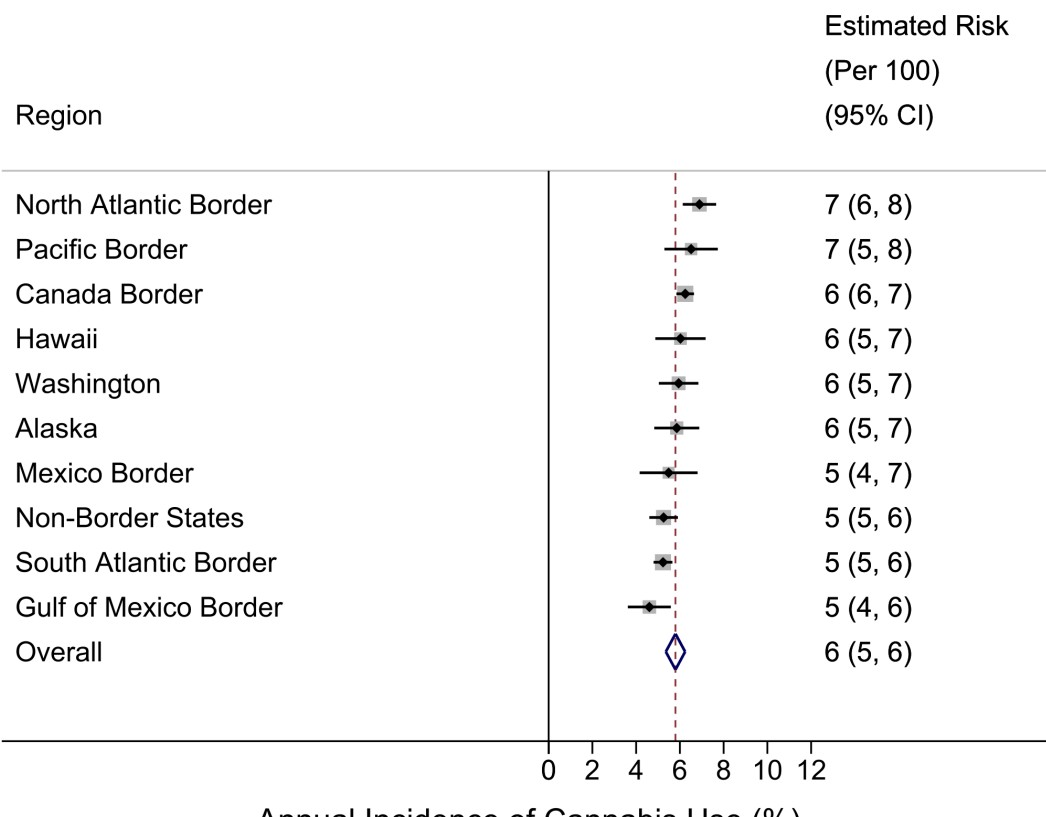

**Figure 2  Region-specific annual incidence rate of cannabis use.** Data from United States National Surveys on Drug Use and Health 10-Year Restricted Data Analysis System, 2002–2011 (Unweighted $n \sim 420,000$ 12–24 year olds). Note: reported estimates and their 95% CIs (i.e., numbers in the 'Estimated Risk' column) are rounded to the hundreds decimal point due to relatively small sample sizes in small regions.

In this regard, other macro-environmental factors that vary state by state, such as higher levels of poverty, lower education attainment, more rapid population growth, and marked social inequalities might contribute to variations in incidence rates for cannabis use onset among young people in various regions and states of the US (*Bruhn, 2014*; *Ganster & Lorey, 2008*). In addition, there are state-specific estimates for the *prevalence* of cannabis smoking (*Hughes, Lipari & Williams, 2013*; *Substance Abuse and Mental Health Services Administration (SAMSHA), 2012a*). The published *prevalence* estimates are from surveys of 12-to-17-year-olds and 18-to-25-year olds in 2010–14, and convey estimated state-specific proportions for individuals with any cannabis smoking at any time during the year prior to the date of survey assessment. The observed patterns in these state-specific prevalence estimates have suggested that the US states bordering Mexico might be remarkable for excess prevalence, with the possible exception of Texas, and have helped substantiate predictions made by others (e.g., *Harrison & Kennedy, 1994*).

Limits on the definitiveness of these published prevalence estimates can be noted by observing that '*prevalence*' is a rather complex multi-component parameter governed by (a) forces that influence 'becoming a newly incident user' as well as (b) forces that influence the

'duration' or persistence of use, once drug use has started. This complexity of prevalence as a summary population statistic in health research is a basic epidemiological principle noted more than 50 years ago by Lapouse, among others. In her work, Lapouse drew attention to the limited utility of prevalence estimates when the aim is to study influences on health, for which knowledge of newly incident cases is required. She recommended restricting the use of prevalence estimates to the planning of health services because prevalence conveys an impression of the total caseload facing a community without discriminating whether the cases are chronic or long-sustained versus newly incident (*Lapouse, 1967*). In research on drug use, it is correct to say that prevalence of drug use serves as an epidemiological indicator of the theoretical 'market demand' for a drug because the market demand segments include both newly incident users observed in any given year, as well as long-time persistent users with drug onsets in prior years (*Cheng, Cantave & Anthony, 2016*). In this paper, we make no direct comparison of cannabis incidence rates and cannabis prevalence proportions. Nevertheless, for readers who might be interested in the comparison of state-level prevalence and incidence estimates, color gradient 'heat maps' of state-specific cannabis prevalence estimates for the US can be found elsewhere (*Substance Abuse and Mental Health Services Administration (SAMSHA), 2012a*).

In this research report, we focus strictly upon the incidence rate for becoming a newly incident cannabis user (i.e., with focus on individuals who started using cannabis for the first time during the year prior to the date of survey assessment). This focus on newly incident users reflects our attempt to ask a more specific research question about regional patterns for observed risk of first time use of cannabis in the US-Mexico border state populations relative to incidence rate estimates for other US regions, and to create state-level benchmark values for use in public health policy and program planning. In an online report based on survey estimates through 2014, Lipari and colleagues (*2015*) recently provided estimates of the number of new cannabis initiates (in millions) and for several age subgroups (e.g., 12-to-17-year-olds), but did not convert these numbers to incidence rates and did not supply state-specific incidence rate estimates, which we present for the first time in the peer-reviewed journal literature.

## MATERIALS AND METHODS

### Study population and samples

For this research, we turned to nationally representative survey samples drawn and assessed each year from 2002 through 2011 for the National Surveys on Drug Use and Health (NSDUH). In these surveys, each state's sample in a given year is a statistically independent replication sample, with state-specific numbers of sample participants guided by the size of the state population. Each year's NSDUH study population is specified to consist of non-institutionalized civilian Americans aged 12+ years residing in the 50 states or the US District of Columbia. Each year, multi-stage area probability sampling approaches encompass residents of non-institutional group quarters (e.g., homeless shelters, college dormitories) as well as residents of household dwelling units. State Sampling Regions (SSRs) are created in each state based on census data. From within SSRs, probability

sampling is used to select the dwelling units (DU) and the rostered inhabitants of each DU, from which a probability sample of designated survey respondents is drawn. The NSDUH field staff are responsible for visiting and securing a roster for each designated DU, and then for introducing and securing informed consent or assent from each designated respondent. The resulting yearly NSDUH samples from 2002 through 2011 have included more than 60,000 designated respondents recruited after informed consent protocols approved by cognizant committees for protection of human subjects ($n{\sim}420,000$ 12-to-24-year-olds). Participation levels have been between 70% and 80% during these years (see supplementary material for a more detailed description). Additional details about NSDUH are provided in online monographs (*Substance Abuse and Mental Health Services Administration (SAMSHA), 2012b*; *US Department of Health and Human Service, 2015*) and many published articles (e.g., *Cheng, Cantave & Anthony, 2016*).

This research project's estimates are based upon the public use datasets created from the NSDUH study operations, which have been termed the Restricted-Data Analysis System (R-DAS), as described in detail in a prior study of state-level estimates for prescription pain-killers (*Vsevolozhskaya & Anthony, 2014*). For this study's cannabis estimates, we turned to R-DAS 10-year datasets for online analyses of the NSDUH 2002–2011 data, with analysis weights crafted for that 10 year interval. Because age-specific incidence of cannabis use drops to extremely low levels at age 24 years, we focused estimation on the subgroup of 12-to-24-year-old young people, yielding an aggregate 10 year unweighted sample size of approximately 420,000 individuals, with no need for age-standardization adjustments given relatively balanced state-by-state age distributions in this age range (*Substance Abuse and Mental Health Services Administration (SAMSHA), 2012b*; *US Department of Health and Human Services, 2015*). (In addition, given state-specific imbalanced distributions at older ages (e.g., population proportion age 50 years and older in the US-Mexico border states versus US-Canada border), the entire age range of NSDUH participants was not considered.)

## Assessment

NSDUH employs audio computer-assisted self-interviews (ACASI) to promote accuracy and completeness of self-reports about drug use and related behaviors. Newly incident cannabis users are identified via a specific module that asks about month and year of first cannabis use, with results recorded in an R-DAS variable called RECMJ_B. Cross-classification of RECMJ_B with the R-DAS variable ELGMJ_B makes it possible to identify individuals who were 'at risk' of starting to use cannabis for the first time during the 12 months prior to assessment, with differentiation of those who did or did not start using during that interval.

The covariates of central interest are US geographic regions crafted by our research team. Prior to analysis, we sorted each state into ten regions pre-specified according to our judgments about potential availability and access to cannabis via neighboring land or water borders. After creating initial regions around the borders of the United States, all remaining states were grouped together into one interior region. Prior to analyses, we re-sorted some states and removed them from border regions as described in the next paragraph.

For our pre-analysis specification of US regions, Alaska and Hawaii were set as 'regions' in and of themselves. All US-Mexico border states (except for California) were placed in one region. California was aggregated with Oregon and Washington State based on the 'Pacific Ocean border' considerations described in our introduction. However, once we considered Washington State's position on the Pacific Ocean and also along the US-Canada border, we placed it into its own single-state region, and placed all other states along the Canadian border in a US-Canada border region (e.g., from Maine to Idaho, including New York State). All other states along the Atlantic Ocean were split into North and South regions based upon where the Chesapeake Bay estuary flows into the Atlantic Ocean. The states along the Gulf of Mexico were aggregated, including Florida (despite its Atlantic shoreline). All remaining states were grouped in a single region as 'interior' or 'non-border' states with neither national borders nor ocean shorelines protected by the US Coast Guard, Customs, or Justice Department agents who now enforce US cannabis regulations as part of Homeland Security protections.

As noted, one goal of these pre-assignments was to address the fact that some states certainly might qualify for membership in multiple regions defined by border considerations. We have a statistical rationale for beginning by placement of each state into only one aggregated region, or leaving it by itself as in the case of Washington State. A resulting statistical advantage is reduced complexity when making incidence rate comparisons across independently specified regions. In this fashion, covariances due to duplicate entries are eliminated.

We ended up with an initial pre-specification of 10 regions (i.e., specified before estimation), as follows: (1) **Alaska**; (2) **Hawaii**; (3) **Mexico Border Region** (Texas, New Mexico, and Arizona; minus California); (4) **Canada Border Region** (Idaho, Montana, North Dakota, Minnesota, Wisconsin, Illinois, Indiana, Michigan, Ohio, Pennsylvania, New York, Vermont, New Hampshire, and Maine; minus Washington State); (5) **North Atlantic Border Region** (Massachusetts, Connecticut, Rhode Island, New Jersey, Delaware, and Maryland); (6) **South Atlantic Border Region** (Virginia, North Carolina, South Carolina, and Georgia); (7) **Gulf of Mexico Border Region** (Florida, Alabama, Mississippi, Louisiana); (8) **Pacific Border Region** (California and Oregon); (9) **Washington State**; (10) **Non-Border States** (Nevada, Colorado, Utah, Wyoming, Nebraska, South Dakota, Oklahoma, Missouri, Kansas, Iowa, Tennessee, Arkansas, Kentucky, West Virginia, and District of Columbia). A small number of controversial assignments have been addressed via post-estimation re-assignments. For example, in a post-estimation analysis we have combined California in a re-assignment with the other US-Mexico border states.

## Analysis

This study's cannabis incidence rate for 12-to-24-year-olds was estimated, state-by-state, as the weighted number of newly incident users divided by the weighted number of 'at risk' individuals (i.e., never users plus newly incident users), based on R-DAS analysis weights for the 10 year interval. We then used meta-analysis to group the states by region and to produce a region-specific summary estimate, with a random effects estimator when heterogeneity chi-square test statistics disclosed a heterogeneity I-squared statistic >0.50

with $p < 0.05$ (*DerSimonian & Laird, 1986*; *Higgins et al., 2003*). Given that state-specific sample sizes are drawn proportional to state population size, all standard errors for state-specific estimates are 'information' weighted. That is, states with larger populations have larger samples. This fact is reflected in the region-wise meta-analysis summary estimates. Finally, a meta-analysis summary estimate for the US as a whole was derived on the basis of the region-specific estimates.

The R-DAS analysis weights for the 10 year interval are used to take into account both sample selection probabilities and post-stratification adjustment factors based upon US Census subpopulation counts. Standard errors and 95% confidence intervals (CI) are from complex survey delta methods. Statistical significance was examined with two indicators that are useful in the large sample context. First, when comparing two independent (mutually exclusive) state-specific or region-specific estimates, an overlap of the two 95% CI usually indicates the null, whereas non-overlap typically suggests a non-null difference with analogous non-null-p-value. Next, we performed formal $z$-test comparisons, following instructions laid out by the NSDUH methods group (*Substance Abuse and Mental Health Services Administration (SAMSHA), 2016*), to assess whether there are any non-null differences between the Mexican border region and other regions that might have been missed using the 95% CIs method. All meta-analyses have been completed using the statistical software Stata Version 13.1 (StataCorp, College Station, Texas, USA).

## RESULTS

Figure 2 shows region-specific estimated incidence rates for becoming a newly incident user of cannabis among 12-to-24-year-old community residents, and a pattern that falsifies our initial expectation. The US-Mexico border region, specified to include Texas, New Mexico, and Arizona, has mid-range cannabis incidence at 5% per year (95% CI [4%–7%]), relative to the US summary estimate of 6% per year. In a post-estimation re-aggregation, we joined California with these three states (given its common Mexico border), and derived a new incidence estimate of 6% (95% CI [5%–7%]), still mid-range (data shown in Data S1). Our conclusion remains the same according to results from the formal $z$-test comparison (Table S3). That is, no differences are found between the US-Mexico border region and each of the other regions.

Figure 2 also shows that, numerically, the largest cannabis incidence rate is seen in the North Atlantic region at 7% per year (95% CI [6%–8%]) and in the Pacific Border region (California plus Oregon) at 7% (95% CI [5%–8%]). In a post-estimation re-specification, we joined Washington State within the Pacific Border aggregate. Relative to the observed range, the resulting new incidence estimate of 6% (95% CI [6%–7%]) remains at the higher end, but it is numerically lower than the 7% estimate for the California-Oregon aggregate (data shown in Data S1).

Readers interested in the state-specific estimates or in re-aggregating the states with different regional specifications may be interested in Figs. 3 and 4, and will find the map's state-specific estimates in our online Supplemental Information. Figure 3 is a heat map, created with the free online software OpenHeatMap, for which each state's color

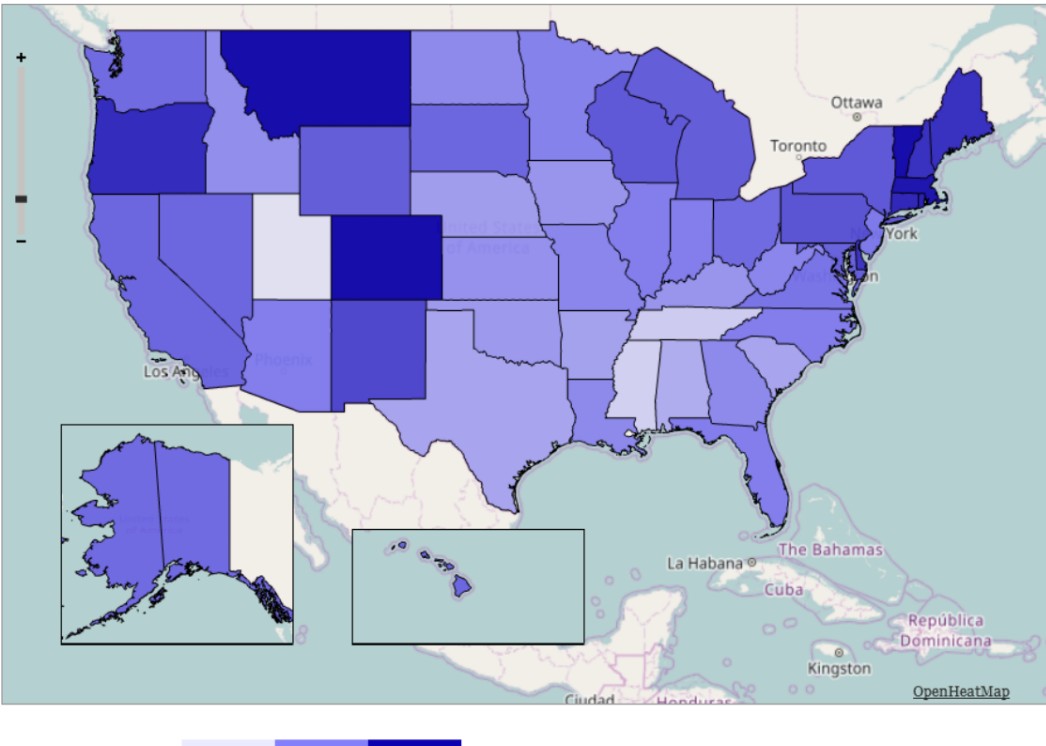

**Figure 3  Heat map representation of state-specific incidence rates.** Gradient heat map representation of each state of the United States in terms of its incidence rate for newly incident cannabis use as estimated for 12-to-24-year-old non-institutionalized civilian community residents. Data from the United States National Survey on Drug Use and Health, 2002–2011. Note: this heat map helps to show that (1) neither the Mexico border states, nor the Gulf of Mexico states have exceptionally large cannabis incidence rates, (2) the northeastern region (including Vermont) has relatively large cannabis incidence rates, and (3) Utah has a relatively small cannabis incidence rate. Comparison of Fig. 3 with Fig. 4 may be useful.

depends on its annual cannabis use incidence estimate, such that the northeastern states are depicted as much darker than the southern states (*Warden, 2010*). Figure 4 is a cartogram or 'blob' map for which each state's relative size is depicted as a function of its annual cannabis use incidence estimate, such that the northeastern states are depicted as much larger than in an area-size depiction. The cartogram was constructed with the mapping software ScapeToad Version 1.2 (*Andrieu, Kaiser & Ourednik, 2008*). The maps and the supplemental state-specific estimates disclose that the 12-to-24-year-old population in Vermont has the largest cannabis incidence rate at 9% per year (95% CI [8%–10%]). The smallest incidence is seen in Utah at 3% per year (95% CI [3%–4%]), shown in Fig. 4 as a small L-shaped state just above the Mexico border states. As described in 'Materials and Methods', Utah was pre-classified as one of our non-border states; its population has some distinctive characteristics that might play a role in this observed lower cannabis incidence estimate, over and above its distance from US borders and coastlines (e.g., see *Vsevolozhskaya & Anthony, 2014*).

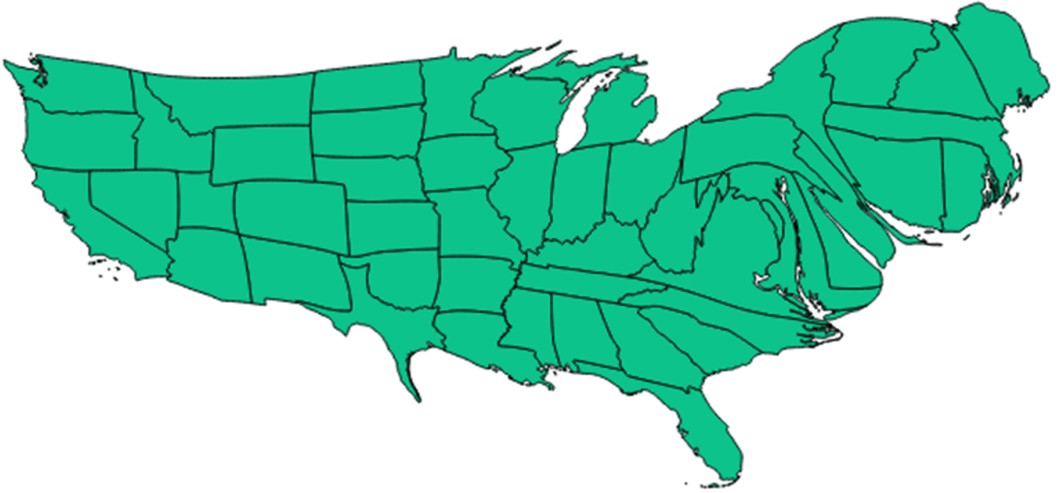

**Figure 4 Cartogram representation of state-specific incidence rates.** Cartogram ('blob map') representation of each state of the United States in terms of relative size of its population incidence rate for newly incident cannabis use as estimated for 12-to-24-year-old non-institutionalized civilian community residents. Data from the United States National Survey on Drug Use and Health, 2002–2011. Note: this cartogram is oriented top (North), bottom (South), left (West), and right (East). It also helps to show that (1) neither the Mexico border states, nor the Gulf of Mexico states have exceptionally large cannabis incidence rates, (2) the northeastern region (including Vermont) has relatively larger cannabis incidence rates, and (3) Utah has a relatively small cannabis incidence rate. (Utah can be located in relation to the Southwest corner of the map where California has a Pacific coastline and shares a southern border with Mexico. California's southeast corner has a border with Arizona, above which is Nevada. Utah can be seen as an L-shaped state just above Arizona and to the East of Nevada.) Comparison of Fig. 4 with Fig. 1 and 3 may be useful.

## DISCUSSION

The most surprising discovery in this research might be the relative homogeneity of the region-specific cannabis incidence estimates within the US. For the most part, all region-specific estimates have overlapping confidence intervals. In consequence, we falsified our expectation that living in relative proximity to the US-Mexico border, nearby the cannabis croplands of Mexico, might produce a larger cannabis incidence rate. The population of the US-Mexico border region has a mid-range cannabis incidence estimate, not appreciably different from those of the other region-specific populations studied here. No clear epicenters or geographical gradients can be seen in the region-wise and state-specific estimates.

All prior epidemiological information about cannabis use in the populations of the US-Mexico border states is based upon regional estimates for prevalence of recently active cannabis use. At present, these states do not appear to have exceptional cannabis incidence rates. If other assertions about elevated cannabis prevalence in this region are correct (e.g., *Harrison & Kennedy, 1994*; *Substance Abuse and Mental Health Services Administration (SAMSHA), 2012b*), but cannabis incidence is not elevated, then the implication is greater persistence or duration of cannabis use once it gets started (*Lapouse, 1967*; *Cheng, Cantave & Anthony, 2016*).

Before any additional discussion of these results, several of the more important study limitations merit attention. The focus on starting to use cannabis in the 12 months prior to assessment constrains the influence of state in- and out-migration on the study estimates, but unrestricted NSDUH datasets included no study variable on the duration of residence in a specific state. For this reason, a small minority of the newly incident users might have started to use in one state, followed by migration into a different state of residence at that time of NSDUH sampling.

Some readers might regard our focus on cannabis incidence rates (becoming a cannabis user for the first time) as a limitation. Here, we return to the issues mentioned in our Abstract and Introduction. Namely, in epidemiological analyses of any condition that involves the agent-host-environment triad, the prevalence proportion mixes up two mechanisms: (1) the mechanisms involved with making effective contact with the agent in the first place, and thereby becoming affected by the agent-attributable condition (in this instance, responding to the chance to try cannabis with actual first-time cannabis use), and (2) the separable mechanisms that determine persistence or duration of the condition after it has started (in this instance, duration of cannabis use). We have a project underway in order to evaluate whether living in the Mexico border region might influence the duration or persistence of cannabis use once it has started, but there are some complexities that limit inferences in that context. One complexity is that we must keep track of how recently the study participant might have moved from a non-border state into a border state, and whether the prevalence of cannabis use might be greater in the border states as a result of seriously involved cannabis users migrating into border states so that they can more readily acquire a supply of cannabis. This situation became more complex after 2011, with an increasing number of non-border states creating a more liberal cannabis regulatory environment (e.g., in Colorado, Oregon, and Washington State) such that migration of long-duration cannabis users to those states might drive up prevalence in the non-border states with liberalized policies, with possibly no effect in incidence rates. We have asked the NSDUH authorities for access to survey information on pre- and post-cannabis migration of the study participants, but our access to those restricted data has not yet been approved.

Another study population issue involves the sampling frame's deliberate exclusion of each state's institutionalized individuals and its military residents. However, this study feature should not have a major influence on estimation of state-specific cannabis incidence rates. Validity and reliability of the survey estimates based upon recalled experience of first cannabis use might vary by state or region (e.g., if influenced by local or state policies governing possession and use, or by law enforcement practices). Nonetheless, we do not consider our findings to be severely biased by this possibility because (a) the recall period is tightly constrained to the most recent 12 months of each participant's lifetime, and (b) the use of ACASI enhances participants' willingness to self-disclose use of internationally regulated drugs (*Substance Abuse and Mental Health Services Administration (SAMSHA), 2012b*). Indeed, the focus on a readily recalled life event (1st cannabis use) within a 12 month time interval of a 12-to-24-year-old participant might represent a major strength of this study as compared to alternative approaches.

Counterbalancing strengths include the study's large nationally representative survey samples with acceptable (although not ideal) participation levels. Whereas cannabis incidence estimates from prospective or longitudinal studies might be considered a gold standard, we note that this study's incidence estimates are free of follow-up attrition biases and involve no measurement reactivity of the type that is faced when an individual is assessed repeatedly in a longitudinal design such that the answers in an earlier assessment can influence answers in a later assessment (*Anthony, 2010*). Finally, when compared to the information value of previous state-specific estimates for cannabis prevalence, this study's use of the incidence parameter constrains potentially confounding influences of in-migration and out-migration after onsets have occurred (*Lapouse, 1967*).

In future research that builds from findings such as these, the role of other local- and region-level factors (e.g., law enforcement, price, 'medical marijuana' policies, etc.) might be explored in order to find the most relevant meso-level correlates with cannabis incidence in young people. For example, notwithstanding concerns expressed by others (e.g., *Manski, Pepper & Petrie, 2001*), there is some research on state- and regional-level variations in price of cannabis products, which could be integrated with incidence analyses (*Ruggeri, 2013*). In addition, both Mexico and Canada have joined the 21st century trend toward relaxation of penalties for cannabis possession and use (e.g., *Freckelton, 2015*). Influences on cannabis incidence in the Mexico and Canada border states may extend beyond relative availability and trade across national borders into the domains of perceptions about normative behavior and whether it is very risky to start using drugs; these perceptions have been found to cluster within neighborhoods of US communities, which is evidence of social sharing that might extend across national and state borders (*Petronis & Anthony, 2000*). Recent estimates suggest that the passage of medical marijuana laws induced null to negative change in the prevalence of cannabis use (*Hasin et al., 2015*); nonetheless, it is possible that a decades-long research process is required to disclose policy impact of this type on prevalence, given that parameter's multi-component complexity as described in our introduction and by *Lapouse (1967)*. Differences in incidence estimates of cannabis use should provide more immediate and possibly more robust evidence for the potential influence of change in cannabis policies of this type. In summary, even though available maps of cannabis flow now draw attention to Mexico's croplands as major sources of supply and availability of this drug within the US (e.g., Fig. 1), the region-specific cannabis incidence estimates of this study falsify our expectation that 12-to-24-year-olds living in Mexico border states might have been more likely to become newly incident cannabis users during the interval from 2002 through 2011. The largest cannabis incidence rate estimates in the United States are not seen in the US-Mexico border region. Rather, it is in Vermont and in the North Atlantic Region that we see the largest estimates.

An increasing number of US jurisdictions are removing penalties for 'recreational use' of cannabis. This study's investigation of geographical variations in cannabis incidence rates creates 'benchmark values' and should provide a useful foundation for a future extension of this line of epidemiological research on 'place' as a potential determinant of starting to use cannabis products. In a recent literature review, *Anthony, Lopez-Quintero & Alshaarawy (2017)* noted a lack of survey-based incidence estimates for cannabis use around the globe

(2017). This study illustrates such estimation of incidence rates using publicly available national data.

State-level public health planners and policy makers might have a more immediate use of this study's estimates, coupled with estimates that roughly 2%-to-4% of young people in the US develop a cannabis dependence syndrome within 12-24 months after first cannabis use (*Chen, O'Brien & Anthony, 2005*). Taking Utah as an example, that state now has an estimated population size of close to 500,000 12-to-24-year-olds who have never tried cannabis (data shown in Table S1). Applying the estimated Utah cannabis annual incidence rate of 3% per year and the published cannabis dependence case transition probability of just 2%, one might project as many as 15,000 newly incident cannabis users in that state population each year, and a total of 300 newly incident cannabis dependence cases possibly needing intervention services each year. (If the 4% transition probability holds, then this caseload count of newly incident cannabis dependence cases grows from a forecast of 300 to 600.) If these forecasts are correct, even when the epidemiology of cannabis use and dependence in Utah is in 'steady state,' with no change from these parameter estimates, the Utah public health officials face a substantial number of newly incident cannabis dependence cases in their population of young people each year. Some of these newly incident cannabis users will require outreach and early intervention services of the type that can be used to reduce person-to-person spread of drug use soon after onset of use in an index user (*Hughes, Lipari & Williams, 2013*). Applied in this fashion, state-level incidence and case transition probability estimates from epidemiological studies become more than 'weather report' statistics or 'yesterday's news' and can add valuable guidance for practical public health planning activities, as well as for evaluation of the impact of future cannabis policy changes.

## ACKNOWLEDGEMENTS

We are grateful to Zachary Sadler and Adnan Moustapha Barazi for their contributions to this project.

### Funding

This project was funded by Michigan State University, [including the MSU Professorial Assistantship Program (JPL)], and the National Institute of Drug Abuse [NIDA T32 DA021129 (HGC and CL) and K05DA015799 (JCA)]. The funders had no role in study design, data collection and analysis, decision to publish, or preparation of the manuscript. This work does not necessarily represent the official views of the National Institute on Drug Abuse.

### Competing Interests

The authors declare there are no competing interests.

## Author Contributions

- Jacob P. Leinweber conceived and designed the experiments, performed the experiments, analyzed the data, wrote the paper, prepared figures and/or tables, reviewed drafts of the paper.
- Hui G. Cheng analyzed the data, wrote the paper, reviewed drafts of the paper.
- Catalina Lopez-Quintero conceived and designed the experiments, analyzed the data, wrote the paper, reviewed drafts of the paper.
- James C. Anthony conceived and designed the experiments, wrote the paper, reviewed drafts of the paper.

## Data Availability

The raw data has been supplied as a Supplementary File.

## Supplemental Information

Supplemental information for this article can be found online at http://dx.doi.org/10.7717/peerj.3616#supplemental-information.

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
