# Peer review of "Newly incident cannabis use in the United States, 2002–2011: a regional and state level benchmark"

_PeerJ, doi:10.7717/peerj.3616_

## Round 0.1 · original submission · Major Revisions

· Academic Editor

Major Revisions

Both reviewers and myself agree this is an important study. I agree with reviewer 1 that the revisions requested would really help solidify the article. The suggestions are clear and I look forward to seeing the revised article.

Reviewer 1 ·

Basic reporting

Although the writing is clear, I think the paper can be shortened considerably without losing any of the content. Some parts are not concise (in particular in the introduction), sometimes the authors mention non-essential information, sometimes they explain the obvious…

Figure 1 is not of high quality. Also, do the authors have permission to reuse it?

Although the existing Figures and Tables are relevant, I would prefer to have a map of the US showing the incidence rates per state (using gradient colours) in addition to (or instead of ) the Cartogram.

Throughout the text the authors are not consistent with punctuation in the in-text citations. For example: “(Manski et al., 2001)” versus “(Manski et al. 2001)”.

The authors refer to various papers/reports with “United States” as author, this should be corrected to include the actual authors/group.

Throughout the text the authors generally report no decimal places for their percentages, but sometimes they do (e.g. line 275). I would personally provide one decimal place throughout the whole paper.

Experimental design

It’s not clear from the methods description how the participants were approached.

The paper would improve if the authors would provide a supplementary table with per state information on the sample size, sex %, age, participation rate etc.

I think the focus of the paper should be on the state-level incidence rates, rather than the subjective regions that the authors focus on now. Also, the one hypothesis the authors have (higher incidence rates in Mexican-border states) seems a bit random (and was not statistically tested).

Given the scope of the paper (becoming a benchmark), and the size of the sample, I think there is a lot more the authors can do with the data. What about sex-specific incidence rates? And why not document and compare incidence and prevalence rates? Why only focus on this particular group, and not also include the 24+ group (incidence may be low, but it may still differ between states). The argument that the population differs between states doesn’t have to be a problem if age is corrected for.

I miss any statistical tests to determine whether incidence prevalence differ significantly between regions/states

Validity of the findings

The results should be compared with existing information on prevalence rates per state.

Any comparison with global figures may also be interesting. And what about any previous incidence estimates in the US? These are not mentioned anywhere in the paper.

Line 283: “All point estimates are within a range of 5% to 7%”, this is only the case when aggregated over states (in subjective regions), so I would not mention this so explicitly.

Additional comments

The authors use data from an impressive sample (N~420,000) to provide an overview of cannabis incidence rates in the US between 2002 and 2011. I agree with the authors that this paper could be used as a benchmark for future research on cannabis use, and therefore it is an important paper to publish and may be well cited. The topic is interesting and timely, given the current policy changes in cannabis use in the US. The manuscript is clearly written.

Aside from the comments mentioned above, I also have some additional (minor) comments:

In the introduction the authors may also want to include the World Drug report from the United Nations Office on Drugs and Crime.

Line 350: The North Atlantic Region, Vermont in particular, has larger cannabis incidence items THAN …

The background part in the Abstract does not really provide a rationale for the study, but rather a declaration of intents.
Line 244: remove period in front of the citation

A few paragraphs are only two sentences long and could better be incorporated with other paragraphs (eg. Lines 64 to 69, 137 to 147, and 345 to 350.

·

Basic reporting

Very interesting and well and careful conducted study. The manuscript is clearly written and I don't have any comments which could improve the quality of this manuscript.

Experimental design

See above. No comment.

Validity of the findings

This manuscript definitely meets the standards of PeerJ. No further comments.

Additional comments

Very well written manuscript. Congrats. I enjoyed reading it.

---

## Round 0.2 · Minor Revisions

· Academic Editor

Minor Revisions

Again, I cannot express how pleased I am to see these revisions. I do agree with the reviewers comments regarding the issues around Confidence Intervals and think it would be helpful to address this. In regard to the second issue regarding sex differences, although I also think this is important, I also have concerns about how we interpret sex differences in these kinds of data. To conclude, I would lime to see point 1. regarding the Confidence Intervals addressed, the second issue, Point 2. regarding sex differences would add to the manuscript but would not lessen the impact of the manuscript if it were not addressed. I look forward to seeing the manuscript again.

Reviewer 1 ·

Basic reporting

no comment

Experimental design

no comment

Validity of the findings

1.13. I agree with the authors that when 95% Confidence Intervals (CIs) of two independent population don’t overlap there will be a statistically significant difference between these populations at p < .05. However, in case of overlapping CIs there may still be a statistical significant difference between population means. To be sure, we suggest the authors to refer to the articles below and rethink the possibility of a formal statistical test. In any case, their statement regarding overlapping CIs (e.g. line 313) should be at least integrated by mentioning to what extent these CIs overlapped.

Austin, Peter C., and Janet E. Hux. 2002. A brief note on overlapping confidence intervals. Journal of Vascular Surgery 36 (1): 194–95.

Knol, M.J., Pestman, W.R., Grobbee, D.E. 2011. The (mis)use of overlap of confidence intervals to assess effect modification. Eur J Epidemiol 26:253–254.

Additional comments

I commend the authors for their revision of the manuscript; overall the work has consistently improved since the first version. I appreciate the observable effort put into the revision work, and particularly the extent to which the authors have been willing to articulate their rationale, as well ast the additional information provided based on the previous comments. I have however two further comments I would like to articulate, which the editor may or may not want to consider before publication.

1.10. I understand that the authors are faced with limited time and resources to be spent on the project, and I acknowledge their original intents and the novelty inherent in providing region- and state-specific incidence estimates. However, it seems to me it would be a valuable addition to give sex-specific estimates as we know there are important gender differences in substance use phenotypes, and specifically for cannabis use (see reference below). Moreover presenting and comparing incidence and prevalence rates between states may consistently extend the scientific impact of the manuscript, especially because this data is already in possession of the authors, and no other report of this entity is available, at present, in the literature.

Substance Abuse and Mental Health Services Administration (SAMHSA). Results from the 2013 National Survey on Drug Use and Health: Summary of National Findings. Rockville, MD: Substance Abuse and Mental Health Services Administration; 2014. HHS Publication No. (SMA) 14-4863. NSDUH Series H-48.

---

## Round 0.3 · accepted · Accept

· Academic Editor

Accept

Thank you so much for your work on this! It has been a pleasure working with you.